# Beyond RAS and BRAF: HER2, a New Actionable Oncotarget in Advanced Colorectal Cancer

**DOI:** 10.3390/ijms22136813

**Published:** 2021-06-24

**Authors:** Chiara Guarini, Teresa Grassi, Gaetano Pezzicoli, Camillo Porta

**Affiliations:** 1Post-Graduate School of Specialization in Medical Oncology, University of Bari ‘Aldo Moro’, 70124 Bari, Italy; pezzicoligaet@gmail.com; 2Division of Medical Oncology, A.O.U. Consorziale Policlinico di Bari, 70124 Bari, Italy; teresa.grassi@libero.it (T.G.); camillo.porta@gmail.com (C.P.); 3Chair of Oncology, Department of Biomedical Sciences and Human Oncology, University of Bari ‘A. Moro’, 70124 Bari, Italy

**Keywords:** colorectal cancer, HER2, targeted therapy

## Abstract

The human epidermal growth factor receptor 2 (HER2) is a well-established oncogenic driver and a successful therapeutic target in several malignancies, such as breast and gastric cancers. HER2 alterations, including amplification and somatic mutations, have also been detected in a small but not negligible subset of patients affected by advanced colorectal cancer (aCRC). However, to date, there are no available oncotargets in this malignancy beyond RAS and BRAF that are available. Here we present an overview on the present predictive and prognostic role of HER2 expression in aCRC, as well as on its consequent potential therapeutic implications from preclinical investigations towards ongoing trials testing anti-HER2 agents in aCRC. While HER2′s role as a molecular predictive biomarker for anti-EGFR therapies in CRC is recognized, HER2 prognostic value remains controversial. Moreover, thanks to the impressive and growing body of clinical evidence, HER2 is strongly emerging as a new potential actionable oncotarget in aCRC. In conclusion, in the foreseeable future, HER2-targeted therapeutic strategies may integrate the algorithm of aCRC treatment towards an increasingly tailored therapeutic approach to this disease.

## 1. Introduction

The treatment of patients with advanced colorectal cancer (aCRC) still relies on the administration of systemic therapies (chemotherapy combined with either antiangiogenics or anti-EGFR agents) with a mainly palliative intent [1]. Recently, clear benefits of a multidisciplinary approach, such as combinations of surgical resection and other locoregional techniques (with or without systemic treatments), have been established for a oligometastatic disease [2,3]. Furthermore, in the recent past, immunotherapy has emerged as a practice-changing treatment [4,5,6,7,8], with Pembrolizumab having received the US Food and Drug Administration (FDA) approval as an up-front treatment for unresectable or metastatic CRC (mCRC) with high microsatellite instability (MSI-H) or mismatched repair deficiency (dMMR) [9].

In the complex treatment decision-making algorithm, patient-related and/or tumor-related characteristics should be taken into account [10,11]. Among patient-related features, clinicians should consider age, performance status, comorbidities, patients’ preferences and values as well as life expectancy [12,13,14], while among tumor-related characteristics, primary tumor sidedness (i.e., right vs. left colon) [15,16,17,18] and molecular profile (i.e., KRAS-mutated vs. KRAS wild-type) [19,20] play a key (and growing) role. Currently, frontline chemotherapy regimens (i.e., the backbone of all available combinations) include leucovorin-modulated 5-fluorouracil, oxaliplatin, and irinotecan agents, which can be associated with monoclonal antibodies (MoAb) directed against the vascular endothelial growth factor (VEGF; bevacizumab) or the epidermal growth factor receptor (EGFR; panitumumab and cetuximab) [2,21]. *RAS* and *BRAF* are the two main genes involved in the EGFR intracellular signaling pathway and, to date, they are the only available biomarkers for aCRC that are routinely used in clinical practice [22,23]. *RAS* mutations, most frequently in exons 2, 3, and 4 of KRAS and NRAS as well as the mutually exclusive *BRAF* ones, lead to the constitutive activation of EGFR downstream transducers and bypasses the EGFR signaling blockade, with a consequent negative predictive role to anti-EGFR target therapy [24,25,26,27,28,29,30,31,32]. Moreover, the BRAF v600E mutation, which is most commonly located in right colon tumors, possesses a recognized unfavorable prognostic value [33,34,35,36]. As a consequence, the use of anti-EGFR agents, although limited to KRAS and BRAF wild type cancers, has contributed to greatly improving the outcome of patients harboring these alterations with a median overall survival (OS) reaching around 30 months [37,38]. However, the development of primary or secondary resistance is almost ineluctable. CRC, indeed, is a highly molecularly heterogeneous disease even in histologically comparable tumors and additional gene alterations frequently arise and accumulate, ultimately leading to disease progression under therapeutic pressure [39,40,41,42,43].

CRC biology and its genomic landscape has been extensively studied with the definition of four consensus molecular subtypes (CMSs) in 2015 [44] and the development of next-generation sequencing (NGS) techniques has advanced our understandings of CRC molecular profile [45]. In the commonly named PRESSING (primary resistance in RAS and BRAF wild-type mCRC patients treated with anti-EGFR monoclonal antibodies) panel, Cremolini and colleagues identified several uncommon genomic alterations, including *HER2* amplification/activating mutations, *MET* amplification, *ROS1*/*NTRK1-3*/*RET* rearrangements, *PIK3CA* exon 20, *PTEN*, and *ALK* mutations. Among all of these gene alterations, the human epithelial growth factor receptor 2 (*HER2*) overexpression/amplification and, less frequently, mutations are the most common ones [46].

In this heterogeneous genomic scenario and in the era of precision medicine, despite the low prevalence of its genetic alterations, *HER2* is emerging both as a key driver in CRC, as well as a predictor of benefits from novel targeted therapies [47]. This review aims at providing an overview of *HER2* role in aCRC ranging from its predictive and prognostic relevance to its potential therapeutic implications as a new actionable oncotarget.

## 2. The HER2 Pathway in Tumorigenesis

### 2.1. HER2 Pathway and Its Alterations in Solid Tumors

The HER2/neu oncogene (also known as ErbB2 or p185), which is located on the long arm of human chromosome 17 (17q12), encodes a transmembrane glycoprotein receptor with intrinsic tyrosine kinase activity [48]. In contrast to other HER/EGFR/ERBB family members, HER2 is defined as an orphan receptor because of the lack of an endogenous ligand. Its activation depends on homodimerization or, most frequently, heterodimerization with other EGFR family receptors, in particular HER3 and EGFR, thus resulting in transphosphorylation of tyrosine residues within its cytoplasmic domain [49,50]. The downstream signal transduction pathways, including MAPK and PI3K/AKT/mTOR, results in cellular proliferation and differentiation, the inhibition of apoptosis, and tumor progression [51,52,53,54,55]. Notably, the HER2-HER3 heterodimers generate more potent stimulator signals than homodimers and particularly in initiating the PI3K/AKT pathway, which is one of the principal regulators of cell growth and survival [56,57] Figure 1.

The biological and clinical roles of HER2 activation have been well investigated in the pathogenesis of several malignancies [58,59] and it has been associated with aggressive tumor behavior, poor prognosis, and resistance to chemotherapy as well as with successful anti-HER2 targeted therapies, i.e., having both negative prognostic, as well as positive predictive value [60,61,62,63,64,65,66].

HER2 protein overexpression, usually (but not always) driven by gene amplification, was one of the earliest alterations identified in human cancers. Initially, HER2 oncogene amplification and/or HER2 receptor overexpression were recognized in up to 30% of breast cancers [67] and in many other tumor types, e.g., gastric, esophageal, lung, bladder, ovarian, endometrial, uterine cervix, head and neck, and colorectal cancers. In breast and gastroesophageal cancers, in particular, HER2 amplification distinguished a specific molecular subtype, with consolidated eligibility for anti-HER2 drugs [68,69,70,71,72]. On the other hand, in several tumors such as lung adenocarcinoma, urothelial, and salivary duct carcinoma, studies evaluating HER2 prognostic and therapeutic roles are still ongoing [73,74,75].

Additionally, in addition to HER2 amplification, somatic activating mutations within HER2 itself have recently been identified as drivers of tumorigenesis and had been observed first in non-small-cell lung cancer and then subsequently in a wide variety of cancers [76,77,78,79,80,81,82]. Heterogeneous HER2 mutations, indeed, have been discovered across all HER2 gene exons involving extracellular, transmembrane, or tyrosine kinase cytoplasmic domains, with the consequent activation of downstream signaling pathways even in the presence of a normal gene copy number [83].

### 2.2. HER2 Alterations in aCRC

Although observed less frequently than in other malignancies and also in colorectal adenocarcinoma, HER2 amplification or somatic mutations have been described in the past two decades, with highly different reported rates of positivity (from <1% to >50%) [84,85,86,87,88,89,90,91]. This variability, particularly in HER2 amplification prevalence, may be due to several factors, including cohort heterogeneity and a small study population, antibody clone selection, staining platform, and different scoring systems [86]. For example, higher rates of HER2 positivity were observed in studies assessing not only membranous but also (and above all) cytoplasmic HER2 overexpression [89]. However, cytoplasmic evaluation has an uncertain biological relevance in CRC as well as in breast and gastric cancers, with no recognized role in standard scoring methodologies because of a possible overestimation of true HER2 positivity [91]. Typically, HER2+ CRC cells present a strong lateral membrane staining, while basal membrane staining is not always observed [92]. The Cancer Genome Atlas (TCGA) project detected HER2 alterations in around 7% of patients affected by CRC, especially in RAS and BRAF wild-type tumors [93]. Although controversial, HER2 overexpression has been observed more frequently in high tumor mutational burden and advanced T stage [86], while some studies evidenced a significant discordance between primary tumors and metastases, suggesting the loss of HER2 positivity during disease progression [94,95]. Moreover, HER2 status has been related to sidedness and, more frequently, HER2-amplified primary tumors have been located in the rectum, as well as in the left colon [96,97,98]. As far as HER2 somatic mutations, they have been reported by TCGA project data in about 4% of CRC and sometimes concomitantly with HER2 amplification or alterations of other oncogenes such as RAS, BRAF, and EGFR. Recently, HER2 activating mutations have also been associated with MSI-H tumors [99]. According to Kavuri and colleagues, these mutations, which are often the same ones identified in breast cancer, include V842I, S3210F, L755S, V777L, and L866M [100]. Additionally, in the PRESSING panel another single HER2 mutation, i.e., the substitution G776V on exon 20, was found [46].

## 3. HER2 Status Characterization in aCRC

Unlike breast and gastric cancers, a specific scoring system for defining HER2 positivity in CRC was not been completely established until the introduction of the HERACLES (HER2 Amplification for Colorectal Cancer Enhanced Stratification) criteria. In 2016, Valtorta and colleagues developed and validated the first diagnostic algorithm for HER2 in CRC during the enrollment for the HERACLES study, which is a phase II trial evaluating the combination of two anti-HER2 monoclonal antibodies (trastuzumab and lapatinib) in KRAS wild-type metastatic CRC patients refractory to standard treatment. These criteria, which are similar but more stringent than the accepted and routinely used ones in breast and gastric cancers, are based on the standard methodologies for the assessment of HER2 protein overexpression and gene amplification and are represented by immunochemistry (IHC) and fluorescent or silver in situ hybridization (FISH or SISH), respectively.

According to Valtorta and colleagues, HER2 positivity is defined as 3+ IHC score (intense expression) in more than 50% of tumor cells or 3+ IHC score in 10–50% of the tumor cells and further FISH confirmation by HER2: CEP17 (chromosome enumeration probe) ratio ≥2 in more than 50% of tumor cells or 2+ IHC score (moderate expression) in more than 50% of tumor cells associated with a FISH positive evaluation. Notably, only membranous HER2 expression in its lateral, basolateral, or circumferential patterns counts toward positivity [92,101,102].

However, other studies applied the HER2 diagnostic criteria proposed by Ruschoff and colleagues in gastroesophageal adenocarcinoma (GEA criteria) and also in CRC [87,89,103,104]. The main difference from the HERACLES criteria consists of the cutoff value for IHC definition. In the GEA criteria, indeed, HER2 positive tumors are those possessing an IHC score of 3+ in more than 10% of tumor cells or an IHC score of 2+ in more than 10% of tumor cells with confirmed HER2 gene amplification by FISH.

Liu and colleagues compared these two different scoring systems in a large cohort of Chinese CRC patients, demonstrating a very high rate of concordance between them despite HER2′s particularly low prevalence in this population (2.9% according to the GEA criteria versus 2.6% according to the HERACLES criteria) [105,106] (Table 1).

Additional approaches to detect HER2 gene amplification, as well as its activating mutations in CRC, include molecular techniques such as next-generation sequencing (NGS) or comprehensive genomic sequencing (CGS). Both of these techniques, which involve profiling the DNA extracted from formalin-fixed and paraffin-embedded (FFPE) tumor samples, allow the identification of HER2 copy number and/or sequence alterations. The strong concordance with gold standard methods (IHC and FISH) across different tumor types supports their application in CRC [107,108]. In the MyPathway phase II basket trial, the efficacy of the two anti-HER2 agents, pertuzumab and trastuzumab, in patients with refractory metastatic HER2 positive CRC and HER2 status have been evaluated not only by IHC and FISH but also by NGS [109].

Recently, an international project among three groups (GI-SCREEN-Japan, NCTN-SWOG-USA, and Korea) aimed at harmonizing the diagnostic criteria for HER2 in metastatic CRC by matching IHC, FISH, and NGS data was launched. Based on several clinical trials testing HER2-targeted therapy, Fujii and colleagues integrated the results of HER2 assessment by IHC/FISH with NGS evaluation across different platforms. After demonstrating the accuracy of IHC/FISH scoring systems and cross-validation of NGS panels, the authors proposed new harmonized HER2 diagnostic criteria in CRC as follows: IHC 3+ score or IHC2+ score associated with HER2: CEP17 ratio > or =2 by FISH in more than 10% of tumor cells for surgically resected samples. In the balance of the risk-versus-benefit ratio, the cutoff of 10% for tumor content had been accepted to allow patient registration in the clinical trials of each country. However, in the efforts to distinguish patients who really would benefit from anti-HER2 agents, they emphasized the importance of performing HER2 assessment on surgical specimens rather than on biopsy ones in order to avoid HER2 false positivity due to HER2 heterogeneous expression in CRC cells [110]. Although this study needs to be validated within a prospective clinical trial, it revealed the opportunity for formulating an international and integrated HER2 diagnostic criteria in CRC, using NGS as a bridge between the two gold methods IHC and FISH. A previous study, conducted by Shimada and colleagues showed that CGS has the same sensitivity as IHC and FISH for recognizing HER2 positive CRC patients who are candidates for anti-HER2 targeted therapy [108]. As a consequence, thanks to its reproducibility and ability to detect gene mutations and copy number variations in a single assay, CGS can potentially facilitate tailor-made treatments.

Moreover, the use of the emerging liquid biopsy even to identify HER2 status in CRC from circulating tumor DNA (ctDNA) represents an attractive future diagnostic tool [111,112,113], but it requires further investigations.

## 4. HER2 Predictive Role to EGFR-Targeted Therapy in aCRC

Since 2011, HER2 alterations have been proposed as a mechanism for both de novo and acquired resistance to EGFR-targeted therapies in aCRC. The first proof of the HER2 role as a potential negative biomarker of response to anti-EGFR agents derived from preclinical data in colon cancer cell lines and xenograft models. In a large platform of patient-derived tumor xenografts (PDTX), Bertotti and colleagues showed a significantly higher prevalence of HER2 gene amplification in cetuximab-resistant RAS, BRAF, and PIK3CA wild-type tumors [114]. Yonesaka and colleagues also investigated the predictive impact of HER2 amplification in metastatic CRC cetuximab-resistant cell lines in vitro and in vivo [115]. These authors demonstrated that HER2 activation, either through HER2 amplification or the HER3-activating ligand heregulin upregulation, results in persistent ERK1/2 signaling and results in anti-EGFR resistance. In addition, the subsequent inhibition of HER2 overexpression by a small interfering RNA, as well as the disruption of the HER2/HER3 heterodimerization by depletion of heregulin, restores sensitivity to cetuximab. Consistent with preclinical observations, they extended these findings in a cohort of 233 cetuximab-treated CRC patients, demonstrating a shorter PFS and OS in those patients with HER2-amplified tumors and higher serum heregulin levels. Furthermore, Mohan et al. detected HER2 amplification in tumors or ctDNA of CRC patients that are non-responding to anti-EGFR MoAb [116]. In the effort to clinically validate HER2 negative impacts on the efficacy of anti-EGFR therapies, other retrospective analyses confirmed the results of Yonesaka and colleagues, evidencing a significantly worse median PFS and OS in patients with HER2 amplification compared with the non-amplification group [117,118,119]. In particular, as suggested by previous preclinical data, Sawada and colleagues revealed that the PFS of HER2-amplificated mCRC treated with anti-EGFR targeted therapy was not only poorer than that of the wild-type RAS/BRAF but also similar to that of patients harboring RAS mutations [118]. In the HERACLES-A study, the OS between patients with HER2-positive CRC and the control cohort was not significantly different. However, the authors of the study reported a lower objective response rate (ORR) to anti-EGFR therapies in HER2-amplified CRC patients and relied on a strong biological rationale, as HER2 overexpression represents an alternative pathway to obviate EGFR signaling in tumorigenesis and/or in tumor progression [101,120].

Similarly, HER2 somatic mutations have also been implicated in the resistance to anti-EGFR agents. Within the frame of the HERACLES project as well as in the PRESSING panel created by Morano and colleagues, the investigators uncovered HER2 activating mutations as an intrinsic predictor of non-responsiveness to the anti-EGFR therapies [101,121].

In the recent past, Pietrantronio and colleagues conducted the first small prospective study to clarify the mechanisms of acquired resistance to anti-EGFR MoAb in mCRC patients. Analyzing both tissue and liquid biopsy samples, the authors illustrated the complex landscape of CRC molecular heterogeneity and detected HER2 amplification within the different and often co-occurrent mechanisms driving secondary resistance to EGFR blockade and converging on MAPK pathway reactivation [122]. Even if currently available data require further validation in larger prospective trials, they all strongly suggest that HER2 represents a relevant molecular predictive biomarker for anti-EGFR therapies in CRC.

## 5. HER2 Prognostic Role in aCRC

In contrast to the higher recognized predictive role for anti-EGFR targeted therapies, HER2 prognostic significance in aCRC is still uncertain. Although early studies [123,124], which are those that considered the cytoplasmic expression of HER2 as a criterion for positivity, reported worse outcomes, the more recent trials [87,118,125,126,127] that only evaluated membranous HER2 expression have not demonstrated a clear correlation with prognosis. In a pooled analysis from three clinical trials (QUASAR, FOCUS, and PICCOLO), Richman SD and colleagues did not find a statistically significant association between HER2 expression and OS [87]. However, Ingold Heppner B and colleagues, in one of the largest cohorts of CRC (from all CRC stages), identified a trend towards a lower OS in patients with HER2-amplified CRC [86]. Subsequently, in another study, which is the the PETACC-8 trial that included exclusively stage III CRC patients treated with adjuvant FOLFOX, revealed an association of HER2 alterations possessing shorter time to recurrence and OS, even after adjustments for multiple clinical and pathological factors [128]. The potential prognostic impact of HER2 is probably hindered by its relatively low incidence in CRC, even if a meta-analysis of 18 studies with 2867 CRC patients confirmed HER2 as an insignificant predictor of survival [129]. Recently, Khelwatty and colleagues evaluated how EGFR and membranous HER2 co-expression impact the outcome of cetuximab-treated patients with mCRC and shoed that when HER2 and EGFR are localized on the cell wall, a shorter PFS is observed [130].

However, even in evaluating the inconsistent results to date available, the HER2 prognostic role remains controversial and its negative effect on OS in CRC, if any, is probably less relevant, than compared with other molecular alterations, e.g., BRAF v600E.

## 6. Anti-HER2 Therapeutic Strategies in aCRC

### 6.1. Preclinical Investigations and Clinical Evidence on HER2 Therapeutic Role in aCRC

Based on the success of routinely used HER2 targeting agents in breast and gastric malignancies, preclinical investigations tested the potential therapeutic role of HER2 in aCRC. A proof-of-concept was derived from HER2-amplified cetuximab-resistant CRC xenograft models where the dual EGFR/HER2 inhibition generated long-lasting tumor regression [114]. In particular, while the anti-HER2 agent pertuzumab, when given alone or in association with cetuximab, induced only a negligible delay in tumor growth, the combination of lapatinib (a dual EGFR/HER2 tyrosine kinase inhibitor) and pertuzumab or, at a lesser extent, of lapatinib and cetuximab caused a significant and durable tumor shrinkage. Similarly, in cetuximab-resistant CRC cell lines, the synergic antiproliferative effect of the anti-HER2 and anti-EGFR agent combinations (e.g., trastuzumab plus lapatinib or trastuzumab plus cetuximab) was demonstrated [115,131,132]. Preclinical findings also showed that the growth of colon cell lines harboring HER2 activating mutations could be potently inhibited by the irreversible tyrosine kinase inhibitors (TKI) neratinib and afatinib. Moreover, in HER2-mutated PDTXs, an anti-HER2 monotherapy (with either trastuzumab, neratinib, or lapatinib) delayed tumor growth, whereas a dual anti-HER2 strategy (with either trastuzumab plus neratinib or trastuzumab plus lapatinib) produced durable tumor regression [100]. According to Kloth and colleagues, activating HER2 mutations also indicates the susceptibility to pan-HER2 irreversible inhibitors in Lynch and Lynch-like HER2 mutated CRC cell lines [133].

All these preclinical data built a solid background and a strong rationale for clinical trials targeting HER2 alterations in patients with aCRC and paved the way for the HERACLES project. During the last decade, several small studies assessed HER2 as a therapeutic target in combination with standard chemotherapies or anti-EGFR drugs. The early clinical studies evaluating the association of anti-HER2 MoAb (i.e., trastuzumab and pertuzumab) and cetuximab or cytotoxic agents (i.e., irinotecan, fluorouracil, and oxaliplatin) were prematurely shut down because of the severe overlapping toxicities [134] or poor accrual [135,136]. Subsequently, in a phase I basket trials including patients with HER2-positive refractory solid tumors, none of six CRC patients experienced an objective response to the combination of paclitaxel, interleukin-12, and trastuzumab [137] and only two of the eight CRC patients had a partial response to the association of cetuximab and lapatinib while in the absence of complete tumor regression [138]. However, the small sample size and the design of these studies probably did not allow the recognition of the contribution of anti-HER2 agents to tumor response.

More recently, preclinical observations [114] supported the clinical investigation of “chemotherapy-free” regimens based on the combination of HER2-targeted drugs. The HERACLES-A (HER2 Amplification for Colo-Rectal Cancer Enhanced Stratification) was the first large phase II clinical trial testing weekly trastuzumab plus daily lapatinib in HER2 positive KRAS exon 2 wild type aCRC patients who proved to be refractory to standard-of-care therapy, including cetuximab. Sartore-Bianchi and colleagues presented the promising results of this pivotal study after the screening of 914 patients according to the stringent HERACLES Criteria. Enrolling 27 HER2-positive eligible patients in the study, the authors reported an objective response rate of 30%, with a median duration of response being 9.5 months and a median PFS of 5.2 months [101]. This response rate, together with the good overall toxicity profile of the treatment (no grade-4 or grade-5 adverse events), compared favorably with other treatment options in heavily pretreated mCRC patients. In line with the HERACLES-A results, the MyPathway phase IIa basket trial, by assessing the association of pertuzumab and trastuzumab in pretreated HER2-amplified mCRC patients, further supports the activity of dual HER2 blockade strategy [139,140]. The updated report of the study, indeed, showed an overall response rate (ORR) of 32% in this cohort of patients, with a median PFS and OS of 2.9 and 11.5 months, respectively. Although, unlike HERACLES, the MyPathway study KRAS status was not evaluated as a criterion of eligibility, the ORR was much higher in the KRAS wild-type mCRC tumors and reached 42%, whereas the efficacy of the pertuzumab/trastuzumab combination was not demonstrated in KRAS-mutated ones and had an ORR of 8% [60]. Interestingly, both the HERACLES-A and MyPathway studies evidenced a significant correlation between a high HER2 gene copy number and a longer PFS from the dual blockade. Moreover, in an exploratory data analysis of HERACLES-A, Siravegna and colleagues proposed a specific plasmatic HER2 copy number to select patients who would really benefit from HER2-targeted therapies [141]. Within the HERACLES project, the HERACLES-B trial also investigated the efficacy of pertuzumab in association with the antibody-drug conjugate trastuzumab-emtansine (T-DM1) in the same subset of patients. Even if the study did not reach its primary endpoint of ORR, the high disease control rate, which is about 68%, the PFS similar to other anti-HER2 regimens, and the safety profile supports the dual anti-HER2 blockade as a potential therapeutic resource for HER2+ mCRC [142].

### 6.2. Ongoing Trials on Anti-HER2 Targeted Therapies in aCRC

Thanks to the encouraging results of phase II studies including HERACLES and MyPathway, many trials testing several anti-HER2 drugs are currently in progress and available preliminary data show that various therapeutic approaches could be active in HER2-positive mCRC patients. Among different new HER2-related strategies in development, a boosted interest is related to HER2-targeted antibody-drug conjugates (i.e., TDM-1, DS-8201, A166, ZW25, and ZW49), novel TKIs (i.e., tucatinib, sapitinib, neratinib, pyrotinib and poziotinib, and Ceralasertib), and HER2-targeted immunotherapy (i.e., vaccines, donor-derived NK cells, and CAR-T cells) (Figure 2). Moreover, preclinical investigations suggested that combination strategies, including the concomitant inhibition of HER2 and other oncotargets (such as PI3K and MEK), are able to induce colorectal cancer stem cell death, leading to cancer regression in xenograft models [143].

Recently, Siena and Coll. published the results of the DESTINY-CRC01 phase II study which shows promising and durable activity of trastuzumab deruxtecan (DS-8201) in patients with refractory HER2-positive mCRC; notably, this activity was evident even in those patients who had previously received HER2-targeted therapies. In particular, in cohort A (including HER2 IHC 3+ or IHC2+ and ISH-positive mCRC patients), the authors documented the regression of target lesions, as well as lasting responses; this ultimately resulted in PFS and OS benefits [144]. Table 2 describes the main features of the principal ongoing trials on HER2 in aCRC (Table 2).

## 7. Future Perspectives and Conclusions

Despite the improvements in the systemic treatment of aCRC made in the past two decades, to date CRC does not have specific oncotargets beyond RAS and BRAF. However, because of its histological heterogeneity and its genetic dynamic evolution during disease progression and under therapeutic pressure, CRC could represent one of the most fertile grounds for the development of precision oncology and approach. In this context, the identification of novel clinically actionable oncogenic drivers expresses an unsatisfied and urgent need. Although, unlike other malignancies, HER2 alterations are observed only in a small (but not negligible) subset of aCRC, an impressive and growing body of evidence supports HER2 status assessment in patients also affected by CRC. In addition to its already consolidated predictive role to anti-EGFR therapy and the more debated prognostic significance, HER2 could, indeed, be a new targetable alteration in aCRC. The introduction of the HERACLES diagnostic criteria for HER2-positive tumors and the initial efforts to integrate traditional methodologies (IHC and FISH) with arising technologies (NGS and CGS) draw a crucial step forward for the accurate characterization of HER2 status in CRC. Moreover, thanks to the growing availability of specific gene panels, the emerging diagnostic tools (NGS and CGS) and the desirable clinical routine use of liquid biopsy, HER2 assessment should be included in all genetic tests for each CRC patient at the time of diagnosis of advanced disease. Although available data and ongoing clinical trials support HER2 therapeutic role in mostly pretreated aCRC patients, similar positive effects are expected in earlier treatment lines. Ongoing trials are required in order to validate this hypothesis. Additional and ad hoc designed clinical studies investigating different typologies of HER2-directed agents are also required in order to define the optimal treatment sequencing in HER2-positive CRC and the most active and efficacious agents. To date, in absence of randomized data, the inclusion of HER2 status definition in the molecular diagnostic workup of all aCRC patients could allow the speedy referral to clinical HER-related trials independent of previous treatment history. However, the strong underlying biological rationale, preclinical findings, and clinical results support consideration for the conventional clinical approval of anti-HER2 therapies by regulatory agencies in a context of an orphan molecular subgroup of patients. In our opinion, HER2-targeted therapies compare favorably with emerging therapeutic strategies in aCRC, including BRAF-directed therapy and immune checkpoint inhibitors. In the near future, anti-HER2 agents and their combination with other drugs may integrate the algorithm of aCRC treatment towards an increasingly tailored therapeutic approach to this disease.

## Figures and Tables

**Figure 1 ijms-22-06813-f001:**
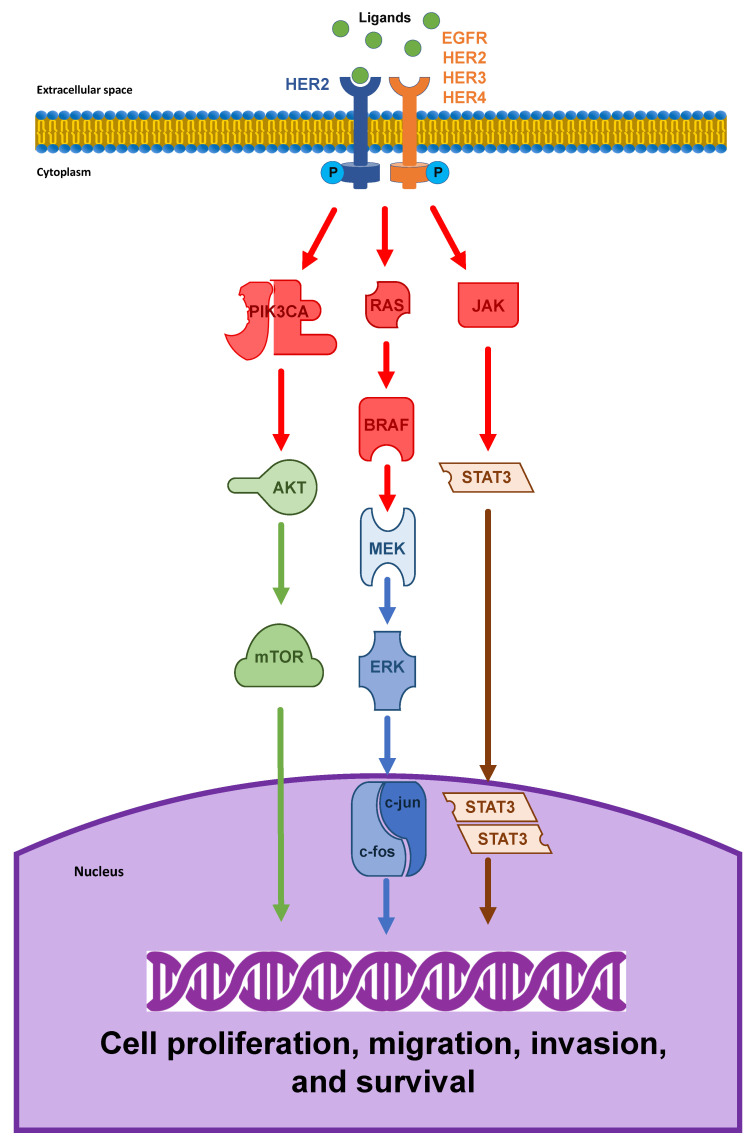
The HER2 pathway in tumorigenesis. Ligand binding to the extracellular domain of human epidermal growth factor receptor (HER2) stabilizes the dimerization between HER2 and another member of EGFR family receptors (EGFR, HER2, HER3, and HER4). By transphosphorylation of tyrosine residues within the cytoplasmic domains, the active homodimers or heterodimers thereafter stimulate several signaling cascades, such as the PI3K/AKT, the RAS/MAPK, and JAK/STAT pathways. These downstream pathways result in the transcription of genes driving tumor cell proliferation, migration, invasion, and survival.

**Figure 2 ijms-22-06813-f002:**
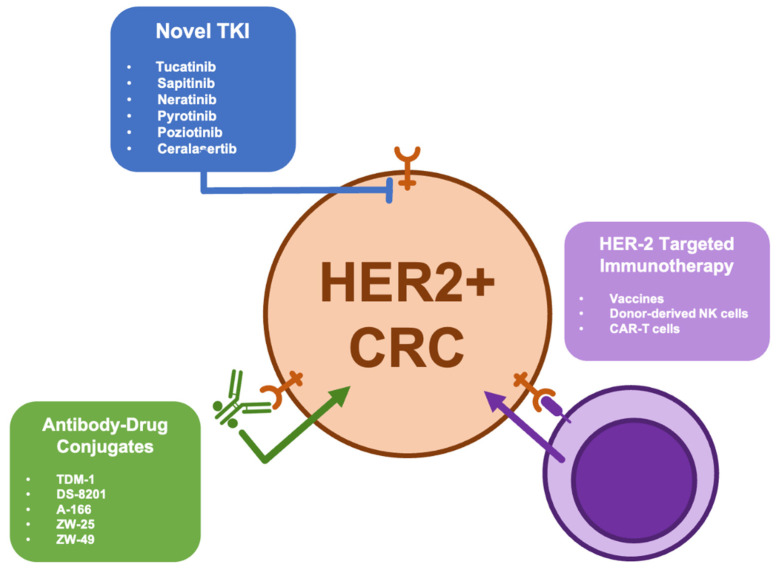
Novel therapeutic strategies for HER-2 positive colorectal cancer.

**Table 1 ijms-22-06813-t001:** The HER2-positivity definition in aCRC according two different system: the HERACLES criteria and the GEA criteria.

HER2 Positivity in aCRC	
The HERACLES Criteria	The GEA Criteria
3+ IHC score * in more than 50% of the tumor cells; OR 3+ IHC score in 10~50% of the tumor cells and FISH positive; OR 2+ IHC score ** in more than 50% of the tumor cells by IHC and FISH positivity ***.	3+ IHC score in more than 10% of the tumor cells; OR 2+ IHC score in more than 10% of the tumor cells and FISH positivity *.

HER2: Human epidermal growth factor receptor 2; HERACLES: HER2 Amplification for Colorectal Cancer Enhanced Stratification; GEA: gastroesophageal adenocarcinoma; IHC: immunochemistry; FISH: fluorescent in situ hybridization. * 3+ IHC score is represented by an intense and strong lateral, basolateral, or circumferential membranous HER2 expression. ** 2+ IHC score is represented by a moderate lateral, basolateral, or circumferential membranous HER2 expression. *** FISH positivity is defined as a HER2:CEP17 (chromosome enumeration probe) ratio ≥2.

**Table 2 ijms-22-06813-t002:** Ongoing currently trials in HER2-positive aCRC patients (clinicalTrials.gov Search Results 05/27/2021; key words: colorectal cancer, HER2).

Title of the Study/NCT	Status/Phase	Intervention	Primary-Outcome Measures
**1. A Study of Pyrotinib Combined with Capecitabine for Metastatic HER-2 Positive Colorectal Cancer/**NCT04227041	Not yet recruiting/Phase I-II	Drug: Pyrotinib in combination with capecitabine	MTD; PFS
**2. A Clinical Study of Pyrotinib in Patients of Advanced Colorectal Cancer with Her2 Variation/**NCT04380012	Recruiting/Phase II	Drug: Pyrotinib Drug: Pyrotinib in combination with trastuzumab	ORR
**3. Study of Neratinib +Trastuzumab or Neratinib + Cetuximab in Patients with KRAS/NRAS/BRAF/PIK3CA Wild-Type Metastatic Colorectal Cancer by HER2 Status/**NCT03457896	Recruiting/Phase II	Drug: Trastuzumab; Drug: Cetuximab Drug: Neratinib Diagnostic Test: Guardant360	PFS
**4. Trastuzumab Deruxtecan in Participants with HER2-overexpressing Advanced or Metastatic Colorectal Cancer/**NCT04744831	Not yet recruiting/Phase II	Drug: DS-8201a 5.4 mg/kg Q3W Drug: DS-8201a 6.4 mg/kg Q3W	ORR
**5. Study of Trastuzumab-emtansine in Patients with HER2- positive Metastatic Colorectal Cancer Progressing after Trastuzumab and Lapatinib (RESCUE)/**NCT03418558	Unknown status/Phase II	Drug: Trastuzumab-emtansine	ORR
**6. Pyrotinib in Combination with Trastuzumab in Treatment- Refractory, HER2-positive Metastatic Colorectal Cancer/**NCT03843749	Recruiting/Not Applicable	Drug: Pyrotinib	ORR
**7. Evaluation of Trastuzumab in Combination with Lapatinib or Pertuzumab in Combination with Trastuzumab-Emtansine to Treat Patients with HER2-positive Metastatic Colorectal Cancer (HERACLES)/**NCT03225937	Unknown status/Phase II	Drug: Trastuzumab, Lapatinib Drug: Pertuzumab, trastuzumab-emtansine	ORR
**8. Monoclonal Antibody Plus Chemotherapy in Treating Patients with Advanced Colorectal Cancer That Overexpresses HER2**/NCT00003995	Completed/Phase II	Biological: trastuzumab Drug: irinotecan hydrochloride	ORR
**9. Tucatinib Plus Trastuzumab in Patients with HER2+ Colorectal Cancer (MOUNTAINEER)/**NCT03043313	Recruiting/Phase II	Drug: Trastuzumab Drug: Tucatinib	cORR
**10. A Phase 1 Study of SHR-A1811 in Patients with Selected HER2 Expressing Tumors/**NCT04513223	Not yet recruiting/Phase I	Drug: SHR-A1811	DLT; RP2D
**11. S1613, Trastuzumab and Pertuzumab or Cetuximab and Irinotecan Hydrochloride in Treating Patients with Locally Advanced or Metastatic HER2/Neu Amplified Colorectal Cancer That Cannot Be Removed by Surgery/**NCT03365882	Recruiting/Phase II	Biological: Cetuximab Drug: Irinotecan Hydrochloride Other: Laboratory Biomarker Analysis Biological: Pertuzumab, Trastuzumab Device: HER-2 testing	PFS
**12. Vaccine Therapy in Treating Patients with Stage IIB, Stage III, or Stage IV Colorectal Cancer/**NCT00091286	Terminated/Early Phase I	Biological: HER-2-neu, CEA peptides, GM- CSF, Montanide ISA-51 vaccine	Safety
**13. Anti-HER2 Therapy in Patients of HER2 Positive Metastatic Carcinoma of Digestive System/**NCT03185988	Recruiting/Phase II	Drug: chemotherapy in combination with trastuzumab for arm1, arm2, arm3, arm 4	ORR
**14. A Study of Poziotinib in Patients with EGFR or HER2 Activating Mutations in Advanced Malignancies/**NCT04172597	Recruiting/Phase II	Drug: Poziotinib Hydrochloride	ORR
**15. Safety and Preliminary Efficacy of SNK01 in Combination with Trastuzumab or Cetuximab in Subjects with Advanced HER2 or EGFR Cancers/**NCT04464967	Not yet recruiting/Phase I-II	Biological: SNK01 Drug: Trastuzumab, Cetuximab	RP2D; ORR
**16. A Clinical Research of CAR T Cells Targeting HER2 Positive Cancer/**NCT02713984	Withdrawn/Phase I-II	Biological: Anti-HER2 CAR-T	CTCAE
**17. Testing the Combination of Two Anti-cancer Drugs, DS-8201a and AZD6738, for the Treatment of Patients With Advanced Solid Tumors Expressing the HER2 Protein or Gene, the DASH trial/**NCT04704661	Not yet recruiting/Phase I	Drug: Ceralasertib Biological: Trastuzumab Deruxtecan	AEs (For Escalation Phase); RP2D
**18. Tucatinib Plus Trastuzumab and Oxaliplatin-based Chemotherapy for HER2+ Gastrointestinal Cancers/**NCT04430738	Recruiting/Phase I-II	Drug: tucatinib Drug: trastuzumab Drug: oxaliplatin Drug: leucovorin Drug: fluorouracil Drug: capecitabine	Renal dose-limiting toxicities; AEs; laboratory abnormalities.
**19. Binary Oncolytic Adenovirus in Combination with HER2- Specific Autologous CAR VST, Advanced HER2 Positive Solid Tumors/**NCT03740256	Recruiting/Phase I	Biological: CAdVEC	DLT
**20. CAR-macrophages for the Treatment of HER2 Overexpressing Solid Tumors/**NCT04660929	Recruiting/Phase I	Biological: CT-0508	Safety and tolerability.
**21. Study of A166 in Patients with Relapsed/Refractory Cancers Expressing HER2 Antigen or Having Amplified HER2 Gene/**NCT03602079	Recruiting/Phase I-II	Drug: A166	MTD
**22. Intravenous TAEK-VAC-HerBy Vaccine Alone and in Combination Treatment in HER2 Cancer Patients/**NCT04246671	Recruiting/Phase I-II	Biological: TAEK-VAC-HerBy	DLT
**23. Phase I/II Trial of Antagonism of HER in GI Cancer/**NCT04246671	Completed/Phase I-II	Drug: AZD8931 Drug: Irinotecan Drug: Folinic Acid Drug: Fluorouracil	DLT
**24. Lapatinib and Cetuximab in Patients with Solid Tumors/**NCT01184482	Completed/Phase I	Drug: cetuximab, lapatinib	MTD
**25. FATE-NK100 as Monotherapy and in Combination with Monoclonal Antibody in Subjects with Advanced Solid Tumors/**NCT03319459	Active, not recruiting/Phase I	Drug: FATE-NK100 Drug: Cetuximab Drug: Trastuzumab	ORR
**26. A Study of SBT6050 Alone and in Combination with Pembrolizumab in Patients with Advanced HER2 Expressing Solid Tumors/**NCT04460456	Recruiting/Phase I	Drug: SBT6050 Drug: pembrolizumab	DLT; AEs
**27. A Dose Finding Study of ZW49 in Patients with HER2-Positive Cancers/**NCT03821233	Recruiting/Phase I	Drug: ZW49	DLT; AEs
**28. ACE1702 in Subjects with Advanced or Metastatic HER2- expressing Solid Tumors/**NCT04319757	Recruiting/Phase I	Drug: ACE1702 Drug: Cyclophosphamide Drug: Fludarabine	DLT; SAEs; MTD
**29. A Safety and Efficacy Study of ZW25 (Zanidatamab) Plus Combination Chemotherapy in HER2-expressing Gastrointestinal Cancers, Including Gastroesophageal Adenocarcinoma, Biliary Tract Cancer, and Colorectal Cancer/**NCT03929666	Recruiting/Phase II	Drug: ZW25 (Zanidatamab) Drug: Capecitabine Drug: Cisplatin Drug: Fluorouracil Drug: Leucovorin Drug: Oxaliplatin Drug: Bevacizumab Drug: Gemcitabine	DLT; CTCAE; ORR
**30. A First-in-human Study Using BDC-1001 in Advanced HER2-Expressing Solid Tumors/**NCT04278144	Recruiting/Phase I-II	Drug: BDC-1001 Drug: Pembrolizumab	SAEs; DLT; MTD; ORR
**31. Study of Bosutinib With Capecitabine In Solid Tumors And Locally Advanced Or Metastatic Breast Cancer/**NCT00959946	Terminated/Phase I-II	Drug: Bosutinib Drug: Capecitabine	MTD; SAEs; ORR
**32. Pembrolizumab and Monoclonal Antibody Therapy in Advanced Cancer/**NCT02318901	Terminated/Phase I-II	Drug: Pembrolizumab Drug: Trastuzumab Drug: ado-trastuzumab emtansine Drug: Cetuximab	RP2D
**33. Targeted Agent Evaluation in Digestive Cancers in China Based on Molecular Characteristics (VISIONARY)/**NCT04584008	Recruiting/Not Applicable	Drug: FGFR Inhibitor, IDH1 Inhibitor, HER2 Inhibitor, PARP Inhibitor, BRAF Inhibitor, MEK Inhibitor, ICIs, EGFR-TKIs, NTRK- TKI, and etc. Drug: Other Therapy	ORR
**34. A Study of T-DXd for the Treatment of Solid Tumors Harboring HER2 Activating Mutations (DPT01)/**NCT04639219	Recruiting/Phase II	Drug: Trastuzumab deruxtecan	ORR
**35. A Study of BDTX-189, an Orally Available Allosteric ErbB Inhibitor, in Patients with Advanced Solid Tumors (MasterKey-01)/**NCT04209465	Recruiting/Phase I-II	Drug: BDTX-189	RP2D; ORR
**36. A Study of SGN-CD228A in Advanced Solid Tumors/**NCT04042480	Recruiting/Phase I	Drug: SGN-CD228A	MTD; SAEs; ORR

AEs: adverse events; CTCAE: Common Toxicity Criteria for Adverse Effects; cORR: confirmed objective response rate; DLT: dose-limiting toxicity; MTD: maximally tolerated Dose; PFS: progression free survival; ORR: objective response rate; RR: response rate; RP2D: recommended Phase 2 dose; SAEs: Serious Adverse Events.

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
