# Peer review of "Beyond RAS and BRAF: HER2, a New Actionable Oncotarget in Advanced Colorectal Cancer"

_ijms, 2021, doi:10.3390/ijms22136813_

Round 1

Reviewer 1 Report

This manuscript submitted as a review by Chiara Guarini, Teresa Grassi, and Camillo Porta covers the predictive and prognostic role of HER2 and other important aspects. This manuscript includes lots of information and the authors have done an excellent job at compelling everything together especially with Table 2. Publication of this manuscript is recommended.

It will be very good to optimize the size of Table 2. There are some minor suggestions:

1) What is the column "outcome" - no information in it.

2) There are blank lines in the 1, 9, 14, 15, and 36 lines.

3) I think you shouldn't write the same words in the column intervention. For example, line 13 can be organized as •Drug: chemotherapy in combination with trastuzumab for arm1, arm 2, arm 3, or arm 4.
4) You can combine the column status and phases. But it is on your consideration.

5) (ORR) - () definition? 

Moreover, 2-3 years' new publications should be presented to better understand the problem that contributes to the area.

It will be very good to present one or two more pictures in the paper according to the information in sections 4-5, and therapeutic strategies 6.

Author Response

Reviewer 1

Point 1: It will be very good to optimize the size of Table 2. There are some minor suggestions:

1) What is the column "outcome" - no information in it.

2) There are blank lines in the 1, 9, 14, 15, and 36 lines.

3) I think you shouldn't write the same words in the column intervention. For example, line 13 can be organized as •Drug: chemotherapy in combination with trastuzumab for arm1, arm 2, arm 3, or arm 4.

4) You can combine the column status and phases. But it is on your consideration.

5) (ORR) - () definition?

Response to point 1: Thanks for the advice. We have now optimized the size of table 2 as requested.

1)and 2) Apologize for the oversight, we have corrected the table as appropriate, and remove blank lines;

3)   and 4) We have modified column status, content, and phases, as suggested;

5)   Apologize for the oversight, we have now defined ORR.

Point 2: 2-3 years' new publications should be presented to better understand the problem that contributes to the area.

Response to point 2: As suggested, we have included 3 more very recent publications concerning HER2 and CRC (see ref. 99, 130, and 143)

Point 3: It will be very good to present one or two more pictures in the paper according to the information in sections 4-5, and therapeutic strategies 6.

Response to point 3: In agreement with the Reviewer’s observation, we made another picture within the therapeutic strategies paragraph (see Fig. 2).

Reviewer 2 Report

This systematic review comprehensively reviews the impact and carcinogenicity of HER2 in aCRC. This review can be used as an important reference for future clinical treatment strategies or drug development and application of aCRC.

Suggestion:

In [1.Introduction] and [7.Future prospects and conclusions], the author can combine shorter paragraphs and condense them into one paragraph for the convenience of readers to read and understand.

In lines 22, 36, 150, and 246, the author can confirm spelling or key in corrections.

Author Response

Reviewer 2

Point 1: In [1.Introduction] and [7.Future prospects and conclusions], the author can combine shorter paragraphs and condense them into one paragraph for the convenience of readers to read and understand.

Response to point 1: Thanks for the advice. We have modified the text, as suggested.

Point 2: In lines 22, 36, 150, and 246, the author can confirm spelling or key in corrections

Response 2: Apologize for the typos; we have now revised those lines, as requested.